

# STTA: enhanced text classification via selective test-time augmentation

Haoyu Xiong[1,*], Xinchun Zhang[1,*], Leixin Yang[1], Yu Xiang[1] and Yaping Zhang[1]

Yunnan Normal University, Kunming, China
* These authors contributed equally to this work.

## ABSTRACT

Test-time augmentation (TTA) is a well-established technique that involves aggregating transformed examples of test inputs during the inference stage. The goal is to enhance model performance and reduce the uncertainty of predictions. Despite its advantages of not requiring additional training or hyperparameter tuning, and being applicable to any existing model, TTA is still in its early stages in the field of NLP. This is partly due to the difficulty of discerning the contribution of different transformed samples, which can negatively impact predictions. In order to address these issues, we propose Selective Test-Time Augmentation, called STTA, which aims to select the most beneficial transformed samples for aggregation by identifying reliable samples. Furthermore, we analyze and empirically verify why TTA is sensitive to some text data augmentation methods and reveal why some data augmentation methods lead to erroneous predictions. Through extensive experiments, we demonstrate that STTA is a simple and effective method that can produce promising results in various text classification tasks.

## INTRODUCTION

Pre-trained language models have demonstrated superior performance on various natural language processing (NLP) tasks (*He, Gao & Chen, 2023*; *Sanh et al., 2019*; *Liu et al., 2019*; *Wang, Wang & Yang, 2022*), leading to a majority of research efforts focusing on improving the performance of the model during training. These efforts include using larger models, various forms of model structures (*He, Gao & Chen, 2023*), adversarial training (*Liu et al., 2020*), and data augmentation (*Fang et al., 2022*; *Ren et al., 2021*). Although some of these methods can bring improvements, they usually require additional training or cumbersome hyperparameter tuning, and even some methods try to obtain negligible improvements at a significant computational cost, which is obviously impractical. These methods attempt to improve the performance of the model from different perspectives during training. However, few studies have focused on improving the performance of the model during inference.

Typically, data augmentation is used during model training by adding transformed copies of each example to expand the dataset. While data augmentation does not require

Corresponding authors
Yu Xiang, xiangyu@ynnu.edu.cn
Yaping Zhang, zhangyp@ynnu.edu.cn

tedious hyperparameter tuning, it often requires additional model training. However, recent work has shown that data augmentation can also be used during inference to obtain greater robustness (*Shanmugam et al., 2020*; *Cohen, Rosenfeld & Kolter, 2019a*), improved accuracy (*Matsunaga et al., 2017a*; *Lyzhov et al., 2020*), or estimates of uncertainty (*Conde et al., 2023*; *Conde & Premebida, 2022*; *Wang et al., 2019*), which is known as test-time augmentation (TTA). TTA is a general method to obtain "smooth" model predictions by aggregating predictions of several transformed versions of a given input. It can be applied to any model and makes no assumptions about the model architecture or training method. For example, predictions of various rotated and scaled versions of a test image can be averaged, so that the final prediction is robust to any single adverse rotation or scaling.

TTA has been widely utilized in computer vision (CV) tasks and has demonstrated remarkable achievements. Previous TTA research mainly concentrate on how to design a better aggregation strategy for the predictions of augmented samples. For instance, *Shanmugam et al. (2020)* proposed to aggregate the augmentation through a learnable neural network, while *Lyzhov et al. (2020)* introduced a greedy policy search method to learn the strategy of data augmentation at test time based on the predictive performance on the validation set. Nevertheless, these TTA methods cannot be directly applied to the text domain and require additional access to the source data. Despite its success in CV, the exploration of TTA's potential in NLP remains nascent. One of the intuitive reasons obstructing TTA research in NLP is the lack of a clear understanding regarding which augmentations should be applied to the text input during testing. Text-based data augmentation methods often involve more drastic transformations compared to image-based augmentations, frequently leading to model mispredictions and diminishing the effectiveness of previous TTA methods. Conversely, in CV, there exists a set of "standard" TTA techniques, such as rotation, scaling, and translation, which are typically perceived as label-preserving and still convey crucial visual information about the depicted object or scene. In contrast, text label-preserving transformations are often task-specific, posing challenges in maintaining intact labels. For instance, methods like words deletion and words position swapping may compromise label preservation.

Learning the choice of augmentation and learning the aggregated augmented predictions usually require access to the source data, which makes new assumptions and introduces additional inference latency, and often incurs unaffordable computational resource costs. As exemplified by the partial-LR method listed in Table 1, it necessitates additional labeled source data for training the learnable network. However, due to the privacy or legal restrictions, such as identity information, patient data, *etc.*, the source data is often inaccessible. Furthermore, due to the lack of effective sample identification and selection mechanisms (*e.g.*, *Mean*, *Max*, *Hard Vote* in Table 1), when facing the augmented samples with large noise generated by unstable text augmentation policies, the aggregated prediction may have a large bias compared to the ground truth. This is often the key obstacle that limits the effective application of TTA in NLP. Despite the popularity of TTA in CV, there is a lack of dedicated research on TTA in NLP. Thus, there is an urgent need for a new TTA method to promote the development of TTA in the NLP field.

**Table 1  The difference between our proposed STTA and related TTA settings.**

| Method | Source free | Online decision | Selective aggregation |
|---|---|---|---|
| Mean | Yes | Yes | No |
| Max | Yes | Yes | No |
| Smote | Yes | Yes | No |
| Hard Vote | Yes | Yes | No |
| Partial-LR | No | No | Yes |
| STTA | Yes | Yes | Yes |

In this work, we mainly focused on: (1) understanding which data transformation version of the prediction changes TTA and, (2) based on these insights, how to reduce the risk of TTA aggregation predictions and further stabilize TTA improvements based on NLP classification. We first empirically analyze common and representative data augmentation methods and discuss their impact on TTA policy design. After the analysis, we propose an online selective TTA method, called "STTA" (Selective Test-Time Augmentation), which divides the transformation samples according to experience and risk. We believe that different versions of the transformed samples contribute differently to the aggregated predictions, so different augmented samples should play different roles during testing. We divide the augmented samples into four roles based on similarity and confidence: gold, reward, potential, and risk. It is important to note that the factors influencing TTA results primarily depend on designing the augmentation policy and effectively aggregating the predictions of augmented samples. In this study, our primary focus is on the latter aspect.

Overall, the research results show that the proposed method is lightweight and easy to implement. Selective TTA can provide significant improvements in the accuracy of text classification and is almost free in terms of computational overhead. Our contributions are as follows:

## Main contributions

- We analyzed and empirically verified why TTA is sensitive to some data augmentation methods and revealed why some data augmentation methods lead to erroneous predictions.
- We proposed a simple yet effective online TTA method, which selectively aggregates augmented predictions based on risk and reward criteria, thus effectively reduces the bias caused by abnormal transformation samples and enhances the robustness of the model.
- Our proposed method is "plug-and-play", can be applied to any existing model without the need for hyperparameter tuning or model modification, and can seamlessly collaborate with other robustness methods.

# EXPERIMENTAL RESULTS

# RELATED WORK

## Test-time augmentation

TTA is a technique applied to a trained model during testing, where multiple augmented samples are generated for each original sample. The average prediction over the augmented samples is then used as the aggregated result to improve the final prediction of the model. Although data augmentation is typically applied during model training, it can also be used during prediction. TTA has been widely demonstrated to enhance model accuracy and robustness (*Krizhevsky, Sutskever & Hinton, 2012*; *Matsunaga et al., 2017b*; *Cohen, Rosenfeld & Kolter, 2019b*), address distribution shift issues (*Zhang, Levine & Finn, 2022*), and defend against adversarial attacks (*Prakash et al., 2018*; *Gao et al., 2020*). Researchers have proposed various TTA methods in different domains, including image segmentation (*Moshkov et al., 2020*), text grammar correction (*Yang et al., 2022*), text classification (*Lu et al., 2022*), audio-text retrieval (*Kim et al., 2022*), theoretical research (*Kimura, 2021*; *Kim, Kim & Kim, 2020*), and uncertainty estimation (*Conde et al., 2023*; *Conde & Premebida, 2022*).

Although TTA has been widely studied in many tasks (*Shanmugam et al., 2020*; *Lyzhov et al., 2020*; *Guo et al., 2017*; *Kimura, 2021*), the fact remains that the source data is assumed to be accessible and there is a lack of in-depth insights into data augmentation policies, which is often impractical in reality. For example, both *Shanmugam et al. (2020)*; *Lyzhov et al. (2020)* utilize a labeled source dataset to learn aggregation. On the other hand, the standard TTA method and the *Max* method select the maximum logit across all augmented samples, and *Smote* (*Fernández et al., 2018*) only considers the nearest sample interpolation to generate new samples, which is prone to interference from noisy augmented samples, resulting in significant bias in aggregation prediction. In this work, we propose a simple yet effective TTA method, called STTA, which effectively combines confidence and similarity to select reliable augmented samples for aggregation.

## Ensembling

Ensemble deep learning models use multiple neural networks instead of a single one to compute predictions to improve the performance of various machine learning problems. Typically, ensembling involves obtaining a set of trained neural network models, each of which has a different algorithm or variant, and averaging the predictions for each test object. The various approaches ranging from traditional methods such as Bagging (*Breiman, 1996*), Boosting (*Schapire, 2003*), Stacking (*Ting & Witten, 1997*), to the latest methods homogeneous ensembles (*Ganaie et al., 2022*) and heterogeneous ensembles (*van Rijn et al., 2018*; *Fang et al., 2021*), have resulted in better performance ensemble models.

## Sub-ensemble selection

Ensembling has been demonstrated to enhance overall performance and robustness. However, training multiple models for ensembling requires additional computational overhead (*Shen, He & Xue, 2019*). Even though a single model is used for TTA, it makes sense to view TTA as an ensemble of different models. This is because each sample of the

augmentation sub-policy in TTA can be regarded as a new test sample. More specifically, multiple models generate multiple different predictions of the same sample, and TTA employs a single model to generate multiple different predictions for multiple augmented samples. When the predictions of multiple augmented samples in TTA are aggregated, it approximates the ensemble of multiple models. A related instance of this concept is found in the work of *Fern & Lin (2008)*, who proposed to select the optimal clustering result from multiple clustering models for aggregation. This demonstrates the notion of TTA and similar aggregations of diverse augmented samples to improve overall prediction accuracy.

## PROBLEM DEFINITION AND MOTIVATION

Let us suppose we have a C-class classification model $f_\theta : x_t \rightarrow \Delta^C$ with parameters $\theta$ trained on labeled source training data $\mathcal{D}_S = \{(x_{t_i}, y_i)\}_i^{N_S}$, where $x_{t_i}$ and $y_i$ represent the input and corresponding label, respectively. During inference at time step $t$, the model can only access the current time step data $x_t$, and the unlabeled target domain data $x_t$ is streamed in a sequential manner. To do this, a batch of $N$ augmented samples $\bar{x} \leftarrow a_i(x) \forall i \in [1, N]$ is generated from a uniform distribution $\mathcal{U}(\mathcal{A})$ of augmentation functions $a \in \mathcal{A}$. More formally, the model $f_\theta$ should be able to make an online decision $\tilde{p}_t$ instead of $p_t$ based on the current input $x_t$ by aggregating the augmented samples $\bar{x}$. Formally, the standard TTA method can be formulated as follows:

$$\tilde{p}_t = \frac{1}{N} \sum_{i=1}^{N} \sigma(f_\theta(\tilde{x}_i)) \tag{1}$$

where $f_\theta(x_i) \in \Delta^C$ and $\Delta^C = \{(p_t^1, \ldots, p_t^C) \in [0,1]^C : \sum_{j=1}^{C} p_t^j = 1\}$ represents a probability simplex. $\sigma : \mathbb{R}^d \rightarrow \Delta^C$ represents the *softmax* function for each $x_t$ to approximate the probability distribution of $p_t$.

### Motivation

As a matter of fact, TTA is still in its infancy in the NLP field, and its effectiveness remains an unresolved issue. It mainly involves two obstacles: (1) It is still unclear which data augmentation methods preserve labels at test time, so simply averaging the predictions of all samples is prone to prediction bias. (2) Previous TTA methods attempt to learn aggregation through neural networks. On the one hand, they assume that the source data is accessible and labeled, as shown in Table 1. On the other hand, previous TTA methods may fail due to inappropriate text augmentation methods. In short, the urgent need has prompted us to propose a selective TTA method.

## WHY SELECTIVELY AGGREGATE AUGMENTED SAMPLES?

To gain a deeper understanding of the reasons underlying the ineffectiveness of TTA caused by text data augmentation, one intuitive approach is to visualize the importance levels of different words in sentences to demonstrate the variations caused by different augmentation strategies. Therefore, we first select the most representative text data augmentation methods, namely *RWD*, *RPI*, *RWS*, *RWI*, and *RWSR*, with detailed information as follows:

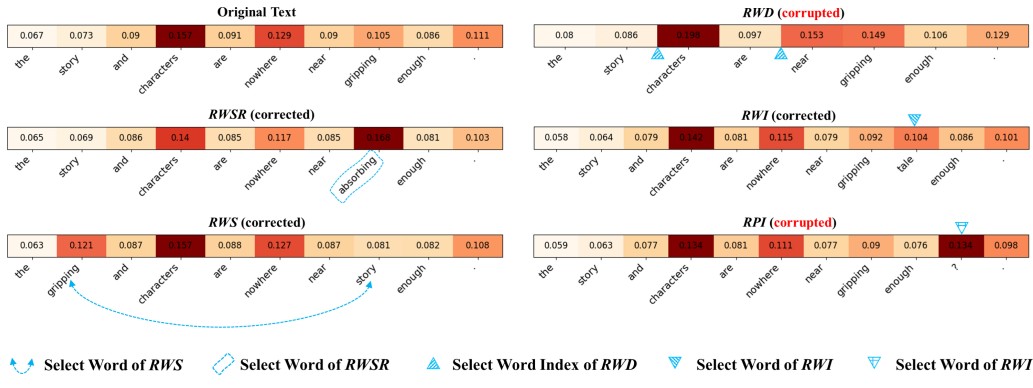

**Figure 1** **Visualization of different data augmentation samples from input $x_t$.** Here, *RWD* denotes the word deletion operation, *RPI* denotes the punctuation insertion operation, *RWS* denotes the word swap operation, *RWI* denotes the word insertion operation, and *RWSR* denotes the word synonym replacement operation.

- *RPI* (Random punctuation insertion) (*Karimi, Rossi & Prati, 2021*): Inserting punctuation marks randomly within a text sequence with a probability of 0.3.
- *RWSR* (Random word synonym Replacement) (*Wei & Zou, 2019*): Randomly replacing *m* non-stop words in a sentence with synonyms with a probability of 0.1.
- *RWI* (Random word insertion) (*Wu et al., 2020*): Inserting a synonym of a randomly selected non-stop word at a random position in the sentence with a probability of 0.1.
- *RWS* (Random word swap) (*Wei & Zou, 2019*): Swapping the positions of two words in a sentence randomly with a probability of 0.1.
- *RWD* (Random word deletion) (*Bayer, Kaufhold & Reuter, 2022*): Randomly deleting words from a sentence with a probability of 0.1.

Following the approach of *Tenney et al. (2020)*, we employed integrated gradients (IG) (*Sundararajan, Taly & Yan, 2017*) to map the importance of different words in the input $x_t$. As shown in Fig. 1, when the input sample is "the story and characters are nowhere near gripping enough," with the true label being "negative," we observed that using the *RWD* method resulted in a transformed sample of "story characters are near gripping enough," involving word deletion and removing the crucial feature "nowhere." This led the model to predict the sample as "very positive," which contradicts the original label. During the application of the *RPI* operation, an error occurred by mistakenly inserting a question mark ("?") at the end of the sentence. Consequently, the meaning of the sentence shifted from a declarative statement to an interrogative one. Although such a change may enhance the semantic richness of the sentence during the training process, it is not suitable for the inference stage.

Thus, if the predictions of all augmented samples are simply aggregated as the final prediction, it will greatly increase the risk of aggregation prediction error, which will lead to a significant drop in the performance of the model. Therefore, it is necessary to selectively aggregate augmented samples.

## METHODOLOGY

In this section, we formally define our proposed method STTA. We first introduce how data augmentation methods are applied at test time. Then we introduce the method of identifying augmented samples based on similarity and confidence score step by step. According to the high and low degrees of confidence and similarity from two perspectives, we can divide the augmented samples into four distinct roles—Gold, Bonus, Potential, and Risk. Based on the identified roles of the augmented samples, we design the selective test-time data augmentation method (STTA). The pseudo-code and overall process for STTA are presented in Algorithm 1 and Fig. 2, respectively.

---

**Algorithm 1** Proposed Approach STTA.

**Input:**    A source pre-trained model $f_\theta$;
           Target domain data $\mathcal{D}_T = \{x_t\}_{t=1}^T$;

**Require:**  Number of augmentations $N$;
           Confidence function $C$; Similarity function $S$;
           Augmentation function $A$;

**Output:** Prediction $\tilde{p}_t \in \Delta_k$ for each $x_t$.

1: **for** $t = 1, \ldots, T$ **do**
2:     Sample $N$ augmented data $\tilde{x}_{t,i} = \{x_t^i\}_{i=1}^N$ using augmentation function $A$.
3:     Feed forward $\tilde{x}_{t,i}$ to obtain the augmented logits matrix $\mathbf{P}^{N \times C}$.
4:     Get the confidence and similarity quartile intervals $Q_{conf}$ and $Q_{sim}$ by Equation 2 and Equation 8.
5:     Get the candidate set $W_{candidates}$ in Equation 9.
6:     Calculate the final prediction probability vector $\tilde{p}_t$ instead of $p_t$ by Equation 12.
7: **end for**

---

### Data augmentation on test time

We first consider $m$ types of augmentation to obtain $N$ augmented samples, where $N = n \times m$ and $n$ represents the number of augmented samples for each type of augmentation. During the inference at time step $t$, for a given test sample $x_t$, we can obtain $N$ augmented samples $\tilde{x}_{t,i}$ by applying $m$ types of augmentation functions $a_i \in \mathcal{A}$ to $x_t$. Then, we can obtain the predicted probability distribution $p_{t,i}$ for each augmented sample $\tilde{x}_{t,i}$ as follows:

$$p_{t,i} = f_\theta(\tilde{x}_{t,i}) \in \Delta^C \tag{2}$$

where $f_\theta(\tilde{x}_{t,i})$ represents the logit associated with the $i$th augmented sample $\tilde{x}_{t,i}$. Then, we can define $\forall i \in [1, n \times m]$,

$$p_{t,i} = [p_{t,i}^1, \ldots, p_{t,i}^C] \in \mathbb{R}^C \tag{3}$$

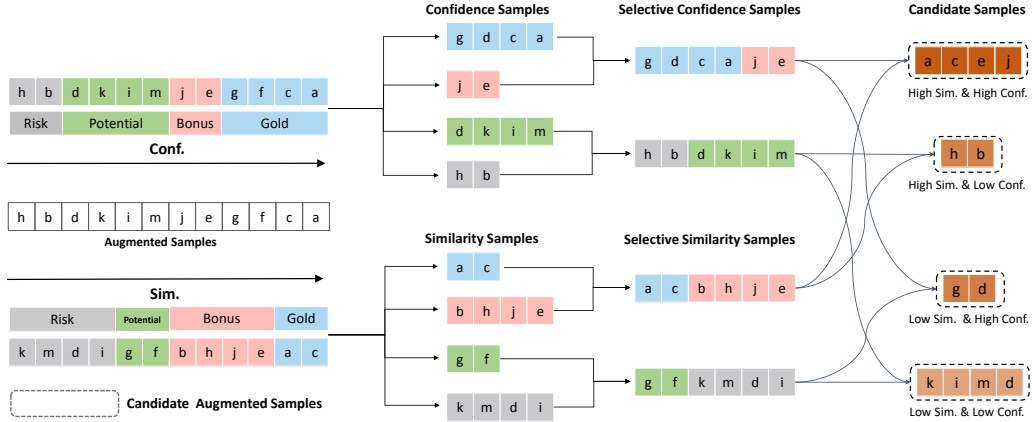

**Figure 2** **An overview of the process of our proposed STTA method.** Different colors represent different augmented samples. The color of the augmented sample is determined by the similarity between the augmented sample and the original sample and the confidence of the model prediction.

and then the augmented logits matrix $\mathbf{P}^{N \times C}$ can be defined as follows:

$$\mathbf{P}^{N \times C} = \left[ p_{t,i} \right]_{i=1,\ldots,N} \begin{bmatrix} p_{t,0}^1 & p_{t,0}^2 & \cdots & p_{t,0}^C \\ p_{t,1}^1 & p_{t,1}^2 & \cdots & p_{t,1}^C \\ \vdots & \vdots & \ddots & \vdots \\ p_{t,N}^1 & p_{t,N}^2 & \cdots & p_{t,N}^C \end{bmatrix} \tag{4}$$

### Similarity-based recognized

Suppose we have a set of augmented samples $\tilde{x}_{t,i}$, where $i \in [1, N]$. Let $g_\phi : \mathcal{X} \to \mathbb{R}^d$ be the model encoder to map the augmented samples $\tilde{x}_{t,i}$ to a $d$-dimensional feature space $\mathbb{R}^d$. Then, we can obtain the feature representation of the augmented samples $\tilde{x}_{t,i}$ as follows:

$$\mathbf{F}^{N \times d} = \left[ g_\phi(\tilde{x}_{t,i}) \right]_{i=1,\ldots,N} \tag{5}$$

where $g_\phi(\tilde{x}_{t,i}) \in \mathbb{R}^d$ represents the feature representation of the $i$th augmented sample $\tilde{x}_{t,i}$.

Then we feed the feature representation $\mathbf{F}^{N \times d}$ into the *HNSW* algorithm (*Malkov & Yashunin, 2018*) to calculate the similarity distance $\mathcal{S}$ between the augmented samples and the original sample as shown in Algorithm 2.

$\mathcal{S}$ indicates the similarity distance between $x_t$ and each augmented sample $\tilde{x}_{t,i}$, so we can divide the augmented samples into quartile intervals by similarity distance.

$$\mathcal{Q}_{sim}(\mathbf{P}^{N \times C}) : \left\{ \bigcup_{q=1}^{4} \mathcal{S}_q \right\} \to \left\{ \mathcal{S}_1, \mathcal{S}_2, \mathcal{S}_3, \mathcal{S}_4 \right\} \tag{6}$$

where $\mathcal{S}_q$ represents the set of augmented samples in the $q$-th quartile interval based on similarity distance.

---

**Algorithm 2** Similarity-based Recognized

---

**Require:** Augmented feature matrix $\mathbf{F}^{N \times d}$

**Ensure:** Similarity distance set $\mathcal{S}$

1: Initialize the graph $G$ with a single node $v_0$ and set $v_0$ as the entry point
2: Current level $l \leftarrow 0$
3: **for** $i = 1$ to $N$ **do**
4:      $v_i \leftarrow HNSW(\mathbf{F}^{N \times d}, v_0)$
5:      Add $v_i$ to $G$ and connect it to $v_0$
6:      **if** $i2^l = 0$ **then**
7:          $l \leftarrow l + 1$
8:      **end if**
9: **end for**
10: Query feature $q \leftarrow g_\phi(x_t)$
11: Initialize the priority queue $PQ$ with $v_0$
12: Initialize the set of visited nodes $V$ with $v_0$
13: $search \leftarrow 0$
14: $efSearch \leftarrow 100$
15: **while** $search < efSearch$ **do**
16:      $v \leftarrow PQ.pop()$
17:      **for** $v_i \in v$ **do**
18:          **if** $v_i \notin V$ **then**
19:              $V \leftarrow V \cup v_i$
20:              $PQ \leftarrow PQ \cup v_i$
21:          **end if**
22:      **end for**
23:      $search \leftarrow search + 1$
24: **end while**
25: $\mathcal{S} \leftarrow \{d(q, v_i) | v_i \in V\}$
26: **return** $\mathcal{S}$

---

## Confidence-based recognized

The confidence of the model prediction is defined as the maximum probability value of the predicted probability distribution of the model. For each augmented sample $\tilde{x}_{t,i}$, the confidence score can be obtained using the following formula:

$$conf(x_{t,i}) = \max_{j=1,\ldots,C} p_{t,i}^j \tag{7}$$

Then, we can obtain the set of confidence scores for augmented samples, $\mathcal{C}$, by sorting the confidence scores of the augmented samples and dividing them into quartile intervals.

$$\mathcal{Q}_{conf}(\mathbf{P}^{N \times C}) : \left\{ \bigcup_{g=1}^{4} \mathcal{C}_g \right\} \rightarrow \left\{ \mathcal{C}_1, \mathcal{C}_2, \mathcal{C}_3, \mathcal{C}_4 \right\} \tag{8}$$

where $\mathcal{C}_g$ represents the set of augmented samples in the $q$-th quartile interval based on confidence score.

## Sample roles recognition

Based on the two perspectives mentioned above, we can divide the all augmented samples into four different roles according to the high and low degrees of confidence and similarity. The details are as follows:

- $W_{\text{gold}}$ refers to samples with high confidence and are the most similar to the original sample.These samples can effectively evaluate and improve the model's generalization ability, making them valuable for aggregation. It can be formulated as $candidates = \{x_{t_i} | x_{t_i} \in (\mathcal{C}_1 \cup \mathcal{C}_2) \cap (\mathcal{S}_1 \cup \mathcal{S}_2)\}$
- $W_{\text{bonus}}$ refers to samples with high confidence and low similarity which can provide additional feature information for aggregation. It can be formulated as $candidates = \{x_{t_i} | x_{t_i} \in (\mathcal{C}_1 \cup \mathcal{C}_2) \cap (\mathcal{S}_3 \cup \mathcal{S}_4)\}$
- $W_{\text{potential}}$ refers to samples with high similarity to the original sample and low confidence which may have some deviation in the augmentation process but still retain the features of the original sample. It can be formulated as $candidates = \{x_{t_i} | x_{t_i} \in (\mathcal{C}_3 \cup \mathcal{C}_4) \cap (\mathcal{S}_1 \cup \mathcal{S}_2)\}$
- $W_{\text{risk}}$ refers to samples with low confidence and similarity to the original sample which lose the features of the original sample and are less valuable for aggregation. It can be formulated as $candidates = \{x_{t_i} | x_{t_i} \in (\mathcal{C}_3 \cup \mathcal{C}_4) \cap (\mathcal{S}_3 \cup \mathcal{S}_4)\}$

Note that although $W_{\text{gold}}$ theoretically has a greater value contribution. However, this does not imply that $W_{\text{risk}}$ and $W_{\text{potential}}$ do not have the ability to contribute.In practical applications, the selection can be made based on specific circumstances. Here, we have selected $W_{\text{gold}}$, $W_{\text{bonus}}$, and $W_{\text{potential}}$ as candidate samples in all experiments.

Base on the above analysis, we can obtain the candidate set $W_{candidates}^K$ for aggregation:

$$W_{candidates}^K = \{x_{t_i} | x_{t_i} \in (W_{\text{gold}} \cup W_{\text{bonus}}) \cup W_{\text{potential}}\} \tag{9}$$

where $K$ is the number of samples in the candidate set, the matrix of logits for the selected samples, denoted as $\mathbf{P}^{K \times C}$, is defined as follows:

$$\mathbf{P}^{N \times C} \rightarrow [AugmentedSamples] \text{Select} \mathbf{P}^{K \times C} = [p_{t,i}]_{i=1,\dots,K} \begin{bmatrix} p_{t,0}^1 & p_{t,0}^2 & \cdots & p_{t,0}^C \\ p_{t,1}^1 & p_{t,1}^2 & \cdots & p_{t,1}^C \\ \vdots & \vdots & \ddots & \vdots \\ p_{t,K}^1 & p_{t,K}^2 & \cdots & p_{t,K}^C \end{bmatrix} \tag{10}$$

The final prediction probability vector $\tilde{p}_t$ instead of $p_t$ is obtained by aggregating the logits matrix $\mathbf{P}^{K \times C}$ and the original prediction probability vector $p_0$ as follows:

$$\tilde{p}_t = \omega \cdot p_{t,0} + (1-\omega) \cdot \frac{1}{K} \sigma(\mathbf{P}^{K \times C}) \tag{11}$$

$$\text{with } \omega = \max \left\{ \omega \in [0, \alpha] : \arg \max_{i \in \{1,\dots,k\}} \tilde{p}_t(\omega) = \arg \max_{i \in \{1,\dots,k\}} p_{t,0} \right\} \tag{12}$$

where $\omega$ is the aggregation weight, $\sigma : \mathbf{P}^{K \times C} \to \Delta_k$ represents the *softmax* function, and $\alpha \leftarrow 0.5$ since we consider both the original prediction and the aggregated prediction equally important. Although more advanced techniques exist to determine the optimal value of $\alpha$. Nonetheless, our primary goal is to demonstrate the ability of our approach to mitigate bias through the memory bank.

# EXPERIMENTS

## Setup

### Benchmark datasets and models

All experiments were conducted on the pre-trained BERT-base (*Devlin et al., 2018*) and obtained the weights of the pre-trained model from Hugging Face Transformers (https://huggingface.co/bert-base-uncased). Then we fine-tuned it on seven benchmark datasets: MRPC (https://www.microsoft.com/en-us/download/details.aspx?id=52398) (*Wang et al., 2018*), Recognizing Textual Entailment (RTE) (https://huggingface.co/datasets/SetFit/rte) (*Wang et al., 2018*), SST-5 (*Socher et al., 2013*) (https://nlp.stanford.edu/sentiment/index.html), TREC-Fine, TREC-Coarse (https://cogcomp.seas.upenn.edu/Data/QA/QC/) (*Li & Roth, 2002*), SUBJ (https://github.com/facebookresearch/SentEval) (*Conneau & Kiela, 2018*), and TweetEval Emoji (https://github.com/cardiffnlp/tweeteval) (*Barbieri et al., 2018*).

### Implementation details

In this article, we focus on the test time, so for the model tuning in the training stage, all our experiments follow the hyperparameter settings provided by the dataset official website. We use the model training code provided by Hugging face Transformers (https://github.com/huggingface/transformers/tree/main/examples/pytorch/text-classification), and adopt Adam (*Kingma & Ba, 2015*) with initial learning rate $2e^{-5}$, batch size 32 and max sequence length 128. During test time, *Partial-LR* adpot Adam optimizer (*Kingma & Ba, 2015*) with a learning rate $2e^{-5}$, *Smote* select the same number of nearest neighbors as the augmented samples as $k$, *Hard Vote* select the most frequent prediction as the final prediction. For our method, no hyperparameter tuning is involved. All experiments were conducted on a single NVIDIA RTX A6000 GPU, and we performed the experiments with five different random seeds.

### Compared methods

In this article, we mainly consider the following strong and representative baselines:

- **Baseline**: Directly use the prediction of the original sample without TTA.
- **Mean** (*Krizhevsky, Sutskever & Hinton, 2017*) (Standard TTA method): Average logit across all augmented samples. This is standard practice in TTA.
- **Max** (Maximum Predicted Probability) (*Guo et al., 2017*): Maximum logit across all augmented samples; This baseline approach involves choosing the prediction with the highest level of confidence from a given set of predictions.
- **Hard Vote** (*Perikos & Hatzilygeroudis, 2016*): Select the most frequent prediction across all augmented samples.

**Table 2  Composition of six different augmentation policies considered in our experiments.**

| Aug. 1: | *RPI* | Aug. 4: | *RWSR* |
|---|---|---|---|
| Aug. 2: | *RWI* | Aug. 5: | *RWS* |
| Aug. 3: | *RWD* | Aug. 6: | *RWSR + RWI + RWS + RWD* |

- **Smote** (Synthetic Minority Over-sampling Technique) (*Fernández et al., 2018*): address class imbalance by synthesizing minority class samples.
- **Partial-LR** (Partial Logistic Regression) (*Shanmugam et al., 2020*): learning $N$ parameters for each augmentation sample to aggregate the prediction.

The Max, Partial-LR and Mean methods are extracted from *Shanmugam et al. (2020)*, we cannot consider all the baselines because some of them are not applicable to text tasks. While these baselines reflect existing work, they are not the only methods for aggregating test-time augmented predictions. Therefore, we further introduce some representative algorithms that can aggregate sample prediction results at test time, such as *Hard Vote* and *Smote*.

## EXPERIMENTAL RESULTS

### Effect of augmentation policies

The factors that affect TTA include data augmentation methods and how to aggregate predictions effectively. In this section, we will conduct extensive discussions on different data augmentation methods to demonstrate the effectiveness of our method.

Note that although more complex data augmentation methods may be more effective in theory, but they are not necessarily representative in practice. Due to the large number of possible combinations of the above data augmentation methods, we mainly consider the most representative data augmentation methods and then combine them to form six data augmentation policies based on insights from *Lu et al. (2022)*, as shown in Table 2, where *RPI*, *RWSR*, *RWI*, *RWS*, and *RWD* represent random punctuation insertion, random word substitution with replacement, random word insertion, random word swap, and random word deletion, respectively. And when the data augmentation method is Aug. 6 (*RWSR + RWI + RWS + RWD*), the standard data augmentation method is equivalent to the TTA method given by *Lu et al. (2022)*.

Figure 3 illustrates a boxplot displaying the distribution of accuracy for different data augmentation policies. It can be observed that our method effectively adapts to various data transformations, even performing well in cases where semantic changes are likely to occur. This indicates that our approach successfully identifies anomalous samples from the batch data, preserving valuable contributions and thereby enhancing the accuracy of the model. In contrast, other methods lacking this capability experience a greater decline in performance when faced with data augmentation policies such as *RWD*.

Additionally, we observe that in other methods, the outliers in the boxplot tend to be located in the region where the model's net gain decreases. However, in our method, the outliers are generally situated in the region where the model's net gain increases. This suggests that our approach has even greater potential to further improve model gains.
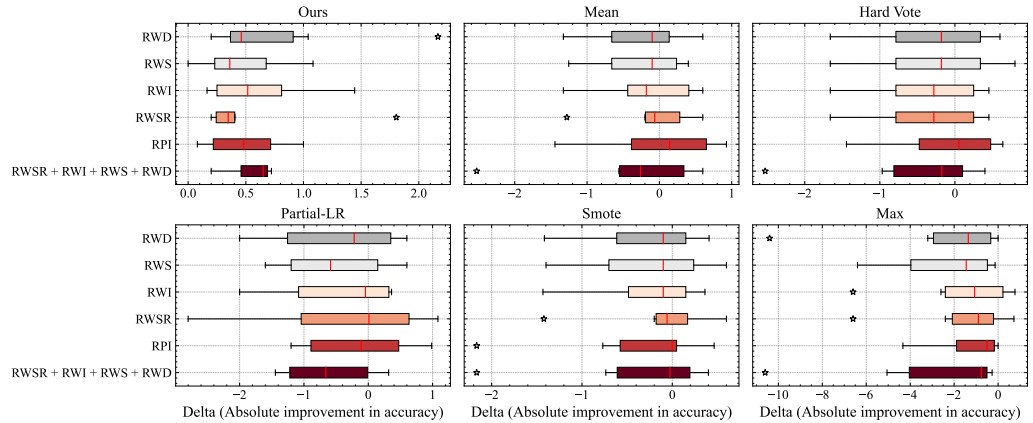

**Figure 3** **Comparison of different data augmentation methods of TTA and report the average accuracy over all datasets.** Here, $N = 32$.

**Table 3** **The absolute accuracy (%) improvement of different TTA methods on 7 benchmark datasets over five random seeds.** Here, data augmentation policy is $RWSR + RWI + RWS + RWD$ and $N = 32$.

| Dataset | SST-5 | MRPC | RTE | SUBJ | TREC coarse | TREC fine | TweetEval Emoji | Avg. |
|---|---|---|---|---|---|---|---|---|
| Baseline | 52.85 | 77.80 | 66.06 | 96.50 | 96.40 | 92.00 | 32.17 | 73.40 |
| Ours | **0.63** | **0.70** | **0.72** | **0.15** | **0.20** | 0.40 | **0.67** | **0.50** |
| Mean | 0.45 | −0.52 | −2.53 | −0.40 | 0.10 | **0.60** | −0.56 | −0.41 |
| Max | −0.50 | −0.52 | −5.05 | −0.45 | −1.00 | −10.60 | −0.27 | −2.63 |
| Smote | 0.18 | −0.23 | −2.17 | −0.40 | 0.20 | 0.40 | −0.74 | −0.39 |
| Hard Vote | 0.14 | −0.35 | −2.53 | −0.40 | 0.00 | 0.40 | −0.97 | −0.53 |
| Partial-LR | 0.32 | −0.64 | −1.44 | −0.35 | −1.40 | 0.20 | −0.68 | −0.57 |

**Notes.**
Results for the proposed method are shown in bold.

Although the gains achieved by our method may not be significant for some data augmentation policies, overall, our approach consistently demonstrates absolute improvements across all employed data augmentation policies.

## Main results

Table 3 displays the absolute accuracy improvement results of all TTA methods on representative benchmark datasets for text classification. In general, most TTA methods fail to generate positive performance gains and, to varying extents, impair the model's performance. Contrarily, our approach achieves significant performance improvements across all datasets, with an average increase of 0.50%. Notably, our approach demonstrates impressive performance gains of 0.72% and 0.70% on RTE and MRPC, respectively, indicating its effectiveness in handling diverse types of datasets.

In contrast, *Partial-LR* utilizes a learnable network to allocate weights to different samples, but it introduces an additional assumption that the source data is accessible. However, its performance improvement falls short of expectations. As discussed in 'Related

Work', the unreasonable data augmentation operations may destroy the sample label, making it challenging for *Partial-LR* to learn the correct weight of each augmented sample.

In the context of uncertain predictions stemming from noisy samples, the baseline model may prove challenging in making robust predictions. Consequently, overconfident and erroneous predictions may arise. Such a scenario presents difficulty in selecting the appropriate high-confidence sample from a set of augmented samples when employing the *Max* method. This, in turn, results in a notable decline of 10.60% in performance on TREC Fine. And *Hard Vote* also faces this obstacle. Furthermore, it is worth noting that while *Smote* achieves the second-best performance on TREC Fine, i.e., 0.40%, its interpolation operation on the original samples may also be impaired by the presence of additional noisy samples. As a result, the problem of erroneous predictions may still persist.

It is worth noting that we observe that when the baseline model performs poorly, such as on the RTE and MRPC datasets, TTA achieves a greater improvement. As the baseline performance improves, when the model performs well, such as SUBJ, TREC Coarse and TREC Fine datasets with an accuracy of over 90%, the improvement of TTA will decrease. Specifically, when we turn our attention to the TREC Coarse dataset, which has a good baseline performance, the improvement of some TTA methods is even 0, such as *Smote*. This finding suggests that TTA is best applied when the baseline model performs poorly. Additionally, *Mean* can achieve the highest performance improvement on TREC Fine with a simple average, but it performs poorly on other datasets, which suggests that there is great potential for TTA, but there is a lack of reasonable ways to identify those noisy samples caused by inappropriate data augmentation methods. Overall, although our method does not significantly improve the datasets on which the model is already performing well, such as SUBJ, our method does not compromise any model performance. These results demonstrate that selectively aggregating predictions based on sample roles is effective.

## ANALYSIS OF LABEL CORRECTION AND CORRUPTION

Figure 4 illustrates the number of samples that have been corrected (model's original incorrect prediction is changed to a correct prediction by TTA) or corrupted (model's original correct prediction is changed to a incorrect prediction by TTA). While the standard TTA method may correct more incorrect predictions on certain datasets (*e.g.*, SST-5), it generally leads to more label corruptions across most datasets. In fact, on some datasets (*e.g.*, TREC Fine), the number of label corruptions is more than double the number of correct labels. This occurs because the standard TTA method fails to consider the distinctions between augmented samples and struggles to accurately determine their significance. Consequently, it often combines some augmented samples with negative impacts alongside other valid samples.

To overcome this disadvantage, *Partial-LR* uses regression methods to estimate the weights of different augmented samples, and then calculate the final prediction results based on these weights. However, it caused the number of corrupted labels to be more than twice the number of corrected labels in SUBJ, TREC Coarse, and TREC Fine datasets. Similarly, *Smote* also caused serious label corruption in SUBJ and TREC Coarse datasets. The reason

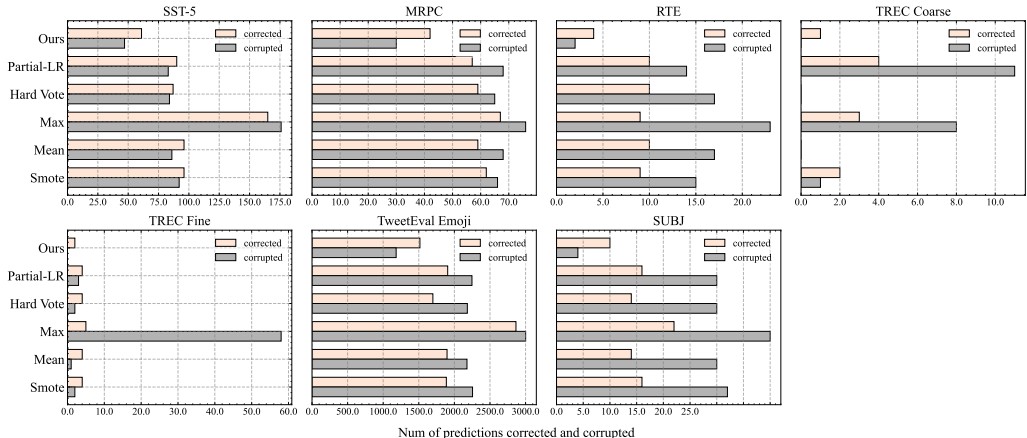

**Figure 4 Number of corrected and corrupted samples by different TTA methods on different datasets.** Here, data augmentation policy is $RWSR + RWI + RWS + RWD$ and $N = 32$.

for the large number of label corruptions is that the $RWSR + RWI + RWS + RWD$ data augmentation policy used in the experiment easily generates abnormal noisy augmented samples. *Partial-LR* and *Smote* are difficult to effectively identify these samples, leading to a greatly increased likelihood of label corruption during the aggregation operation.

To summarize, due to the lack of the ability to distinguish the importance of augmented samples, the model's aggregated prediction results are easily affected by bad augmented samples, resulting in incorrect prediction results.

# ABLATION STUDY

## Necessity of both similarity and confidence

To verify the importance of both confidence and similarity in STTA, we labeled high (low) confidence samples as $C_h(C_l)$ and high (low) similarity samples as $S_h(S_l)$. Then we conducted the following ablation experiments:

- STTA w/o Conf.: When discriminating the roles of the augmented samples, we only use similarity to divide them into two types: $S_h$ and $S_l$.
- STTA w/o Sim.: When discriminating the roles of the augmented samples, we only use confidence scores to divide them into two types: $C_h$ and $C_l$.
- STTA with Conf.&Sim.: When discriminating the roles of the augmented samples, we use both similarity and confidence scores to divide them.

From Fig. 5, the experimental results indicate that conducting experiments solely based on similarity or confidence both lead to a decline in model profit gained, while STTA w/o Conf. is worse than STTA w/o Sim., which indicates that similarity plays a greater role than confidence in aggregating predictions. Furthermore, the results also show that STTA with Conf. & Sim. indicates the best performance, which demonstrates the necessity of combining similarity and confidence to define the different roles of augmented samples.

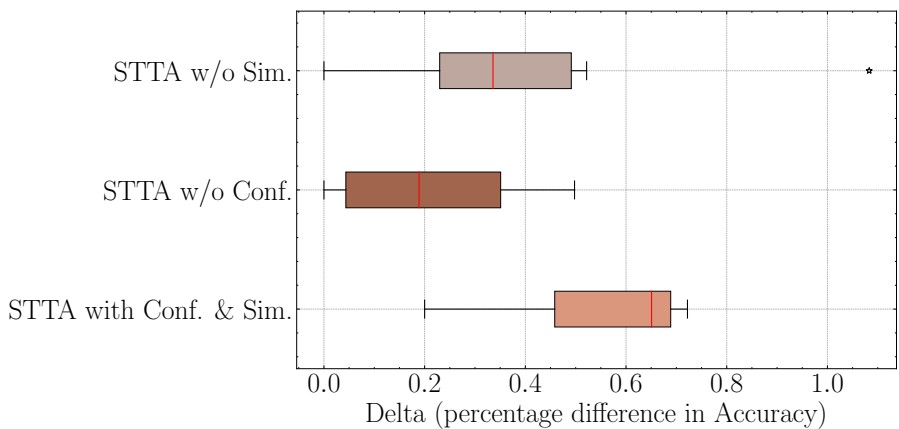

**Figure 5** Necessity of both similarity and confidence. Here, data augmentation policy is $RWSR + RWI + RWS + RWD$ and $N = 32$.

### Effect of different roles

To verify the necessity of different sample roles ($W_{gold}$, $W_{bonus}$, $W_{potential}$ and $W_{risk}$) and the combination of different sample roles, we conducted experiments separately. As shown in Fig. 6, we observed that the gains of $W_{gold}$, $W_{bonus}$, $W_{potential}$ and $W_{risk}$ exhibit a linear downward trend, which confirms the effectiveness of our method in distinguishing valid samples. However, any single sample recognition role is not as good as the method (ours) that combines different roles, which indicates that it is necessary to combine different roles.

### Effect of number of augmentations

Our analysis of the results presented in Fig. 7 aims to explore the impact of different quantities of data augmentation on the performance of TTA. Notably, our proposed method, STTA, exhibits a distinct linear positive correlation between the number of augmentations and model performance. Conversely, the standard TTA method does not yield any discernible advantage with increasing the number of augmentation, while the *Max* method exacerbates the decline in model performance. Although *Partial-LR* mitigates the detrimental effects as the number of augmentations increases, it still falls short of achieving positive gains.

## CONCLUSION AND LIMITATIONS

In this article, we propose a Selective Test-Time Augmentation method, called STTA, which is a simple yet effective alternative to the standard TTA method. We aims to overcome the limitations of the standard TTA method and mitigate the sensitivity of the model to abnormal augmented samples by leveraging the role recognition of augmented samples. Unlike prior advanced TTA methods, STTA does not require access to any source data or additional training. Furthermore, STTA does not interfere with the training process of the backbone network and can be used in conjunction with other robust methods to further enhance the model's performance. Furthermore, our proposed method is straightforward

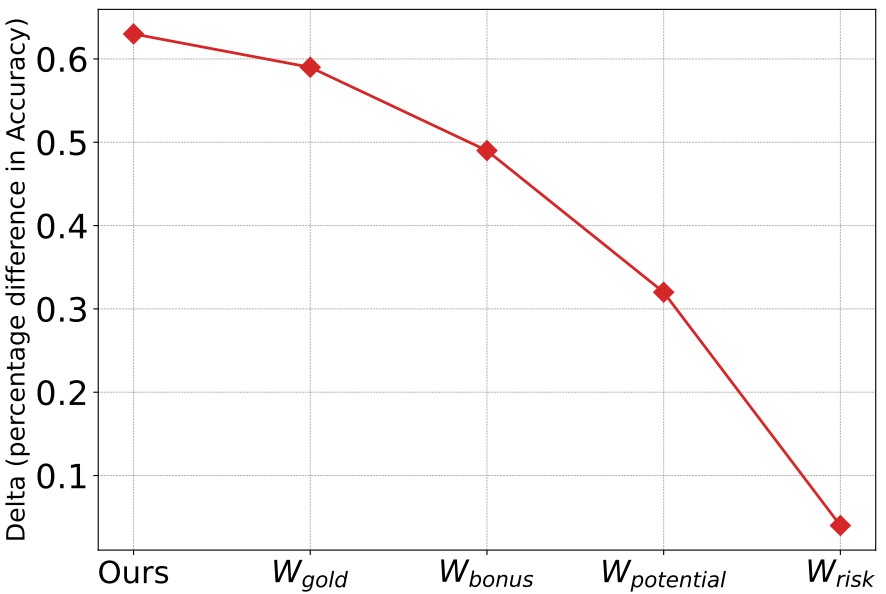

**Figure 6** **Effect of $W_{gold}$, $W_{bonus}$, $W_{potential}$ and $W_{risk}$ on SST-5.** Here, data augmentation policy is $RWSR + RWI + RWS + RWD$ and $N = 32$.

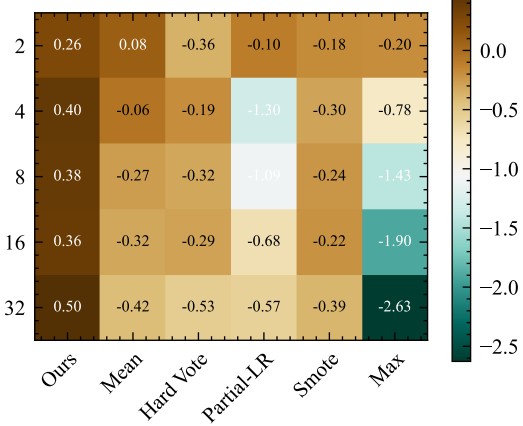

**Figure 7** **Average absolute accuracy (%) improvement of various TTA methods with different numbers of data augmentations is obtained by averaging the results from different datasets and five random seeds.**

and efficient, and its plug-and-play implementation allows for seamless integration with any existing models

For future work, we plan to explore the following directions:

- Investigate how to selectively apply TTA to only those samples that need to be corrected, rather than the entire test set, in order to minimize time and computational costs.

- Design more reasonable and effective data augmentation methods, especially for test time, to further improve model performance.
- Explore the application of STTA in other NLP tasks, such as neural machine translation and question answering, to evaluate its effectiveness and generalizability.
- A potential risk with our method is that the confidence mechanism in Eq. (7) does not fully reflect the reliability of the augmented samples, where the neural networks are overconfident (*Wei et al., 2022*), although we further combine similarity as a judgment. We plan to consider using better methods for uncertainty estimation to evaluate the augmented samples (*Ovadia et al., 2019*).

### Funding
This work was funded by the Ten Thousand Talent Plans for Young Top-notch Talents of Yunnan Province (Project No. YNWR-QNBJ-2018-351). The funders had no role in study design, data collection and analysis, decision to publish, or preparation of the manuscript.

### Grant Disclosures
The following grant information was disclosed by the authors:
The Ten Thousand Talent Plans for Young Top-notch Talents of Yunnan Province: YNWR-QNBJ-2018-351.

### Competing Interests
The authors declare that there are no competing interests.

### Author Contributions
- Haoyu Xiong conceived and designed the experiments, performed the experiments, performed the computation work, authored or reviewed drafts of the article, and approved the final draft.
- Xinchun Zhang conceived and designed the experiments, performed the experiments, analyzed the data, prepared figures and/or tables, authored or reviewed drafts of the article, and approved the final draft.
- Leixin Yang performed the experiments, prepared figures and/or tables, authored or reviewed drafts of the article, and approved the final draft.
- Yu Xiang performed the experiments, analyzed the data, performed the computation work, prepared figures and/or tables, and approved the final draft.
- Yaping Zhang analyzed the data, performed the computation work, prepared figures and/or tables, authored or reviewed drafts of the article, and approved the final draft.

### Data Availability
The code and datasets are available in the Supplementary File.
The original datasets are available at:
- SST-5, Stanford Sentiment Treebank, http://nlp.stanford.edu/sentiment

- MRPC and RTE, Glue benchmark (https://openreview.net/pdf?id=rJ4km2R5t7), https://gluebenchmark.com/
- TREC-Corase and TREC-Fine, The Text REtrieval Conference question dataset (https://trec.nist.gov/), https://huggingface.co/datasets/trec
- TweetEval-emoji, TweetEval benchmark (https://aclanthology.org/2020.findings-emnlp.148.pdf), https://huggingface.co/datasets/tweet_eval
- SUBJ, SentEval (https://github.com/facebookresearch/SentEval), https://huggingface.co/datasets/SetFit/subj

## Supplemental Information

Supplemental information for this article can be found online at http://dx.doi.org/10.7717/peerj-cs.1757#supplemental-information.

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
