# Peer review of "STTA: enhanced text classification via selective test-time augmentation"

_PeerJ Computer Science, doi:10.7717/peerj-cs.1757_

## Round 0.1 · original submission · Major Revisions

Both reviewers are supportive of the quality and novelty of the work but also believe that there are some aspects to improve. Most importantly, they identify areas of the paper, not least the motivation, research questions, and analysis of results, which would need fleshing out to make it clearer.

Please check reviewer comments carefully and provide a point-by-point response to them as you prepare a revision of your work.

·

Basic reporting

See below

Experimental design

See below

Validity of the findings

See below

Additional comments

Overall Evaluation: Accept with Minor Revisions

The manuscript presents Selective Test-Time Augmentation (STTA), a novel approach aiming to enhance text classification by carefully selecting transformed samples for aggregation based on confidence and similarity scores. The study is well-structured, providing a comprehensive introduction to Test-Time Augmentation (TTA) and its challenges in Natural Language Processing (NLP), as well as a detailed description of the proposed method and extensive experiments to validate its effectiveness.

Major Comments:

Introduction:
Strength: The introduction provides a clear context on the significance of Test-time Augmentation (TTA).
Recommendation: A specific citation validating the claim "TTA has been widely demonstrated to enhance model accuracy and robustness, address distribution shift issues, and defend against adversarial attacks" would provide further credibility to the introduction.

Related Work:
Strength: The related work section gives insights into the existing literature and situates the study within that context.
Recommendation:
- The chapter should have a sharper focus on differentiating this study from previous work and/or identifying gaps in the existing literature.
- Ensure that the content isn’t repetitive. For instance, the statement "Ensemble learning is a machine learning strategy that makes decisions based on the predictions of multiple models" appears to be redundant.
- Introduce the methods from Table 1 earlier in the text for better clarity and flow.

Chapter 3:
Strength: Comprehensive coverage of the methods.
Recommendation: It’s essential to describe the methods in the text body rather than relying solely on figure captions.

Methodology:
Strength: The methodology section seems solid, exploring the utilization of confidence scores and similarity in the STTA method.
Recommendation: The problem with confidence values is that neural networks are overconfident (see, e.g., "Mitigating Neural Network Overconfidence with Logit Normalization"). I think it would be best to address this in the limitations of your paper, especially since many researchers say that the values are not reliable ("The issue with many deep neural networks is that, although they tend to perform well for prediction, their estimated predicted probabilities produced by the output of a softmax layer can not reliably be used as the true probabilities (as a confidence for each label)" - https://stats.stackexchange.com/questions/309642/why-is-softmax-output-not-a-good-uncertainty-measure-for-deep-learning-models)

Experiments:
Strength: An in-depth experimentation section, showcasing a comprehensive evaluation setting across diverse tasks and replicable runs.
Recommendation: Start the experimentation section by describing the methods to maintain a logical flow. For instance, the methods mentioned in line 272 about data augmentation strategies should be introduced at the beginning of this section.

Ablation Study:
Strength: The ablation section is well-constructed, providing value to the paper.

Minor Comments:
Citations: Revise the in-text citations for correctness. For example, "HNSW algorithm Malkov and Yashunin (2018)" should be corrected to (Yashunin, 2018) and "integrated gradients (IG) Sundararajan et al. (2017) to map" should be (Sundararajan et al. 2017).
Text Corrections: Address the following issues:
Line 272: Remove the redundant "where."
Line 289: Correct the placeholder "(as described in Section ??)" with the appropriate section number.
Line 301: Consider revising the title "DOES LABEL CHANGED".
- Line 302: "samples that have been corrected (where the model’s original correct prediction is changed to an incorrect prediction by TTA)" -> model's original incorrect prediction is changed to a correct prediction by TTA
Consistency: Ensure uniform notation throughout the paper, especially for terms like SMOTE.
Introductions: The term EDA, first mentioned in line 315, needs an introduction earlier in the text.

Conclusion: The paper showcases promising research on enhancing text classification through selective test-time augmentation. With minor revisions, particularly in citations, content flow, and clarifying specific terms, the paper can achieve a more polished and comprehensive presentation.

Reviewer 2 ·

Basic reporting

-Some of the content is repetitive
- The acronyms need to be elaborated only once and can be used in the rest of the article. Many times they are abbreviated in the article.
- The introduction section started well. Add most relevant works related to TTA or discuss a coherent case for which this research question is important.
- The sections of the article need to tailored again.
- The raw data and the source code are shared.
- Should include more clarity on the results and discussion should be more elaborative.
- The research questions / goals need to be rewritten. They are not clear.
- the difference between test-time augmentation and data augmentation during training is mentioned many times. mention about both of them only once.
- The section heading is not well formatted. (For eg: Related work at line 85 and 86)
- TTA acronym is repeating many times in the manuscript.
- In line 208, the learning rate is misformatted (in 2e-5, -5 is set as superscript)
- The augmentation methods RPI, RWSR, RWI, RWS need to be introduced before their usage in figures.

Experimental design

- The Test-Time Augmentation techniques is not well experimented in the domain of NLP. The work "Selective test-time augmentation" in NLP is interesting as all text augmentation techniques may not generate good examples.
- the methodology is good.
- Clarity is missing in the selection procedure. Is there any threshold set on similarity and confidence to select augmentations or simply top m out of N were selected.
- The Research questions need to be more elaborative.
- The section heading is not well formatted. (For eg: Related work at line 85 and 86)
-Is ensembling of models and and aggregating the predictions same? What is the significance of ensembling with respected to the proposed work?
-From lines 123 to 125 : How can an augmented sample is treated as a model? This claim seems to be inappropriate.
- The problem statement is stated well. But there is no mention about the augmentation selection procedure.

Validity of the findings

The results are well presented. But some of the content is repetitive.
A clear discussion section section is needed. Why performance is dropping in TREC Fine. any reason?

Cite this review as

---

## Round 0.2 · accepted · Accept

Based on the reassessment of one of the original reviewers and myself, I'm happy to confirm that this submission has satisfactorily addressed reviewers' concerns and can therefore be accepted for publication in its present form.

·

Basic reporting

no comment

Experimental design

no comment

Validity of the findings

no comment

Additional comments

After the examination of the revised manuscript and careful consideration of the changes made in response to my comments and those of the other reviewer, I am pleased to report that the authors have satisfactorily addressed all the major concerns. The revisions have significantly improved the manuscript, both in terms of content and presentation.

Clarity and Structure: The authors have successfully clarified the sections, resulting in a much more coherent narrative. The structural changes have enhanced the flow of information, making it easier for readers to follow the argument.

Literature Integration: The revised manuscript now includes a more comprehensive integration of relevant literature and detailed description of the research gap.

Response to Other Reviewer's Comments: The authors have appropriately addressed the points raised by the other reviewer. They have provided detailed explanations and justifications where necessary, and have made changes that reflect a careful consideration of those comments.

Given the thorough and thoughtful revisions made by the authors, addressing the concerns raised in the initial review process, I recommend the acceptance of this paper for publication.